# The Stress-Inducible BCL2A1 Is Required for Ovarian Cancer Metastatic Progression in the Peritoneal Microenvironment

**DOI:** 10.3390/cancers13184577

**Published:** 2021-09-12

**Authors:** Rui Liang, Mingo M. H. Yung, Fangfang He, Peili Jiao, Karen K. L. Chan, Hextan Y. S. Ngan, David W. Chan

**Affiliations:** 1Shenzhen Institute of Research and Innovation (HKU-SIRI), The University of Hong Kong, Shenzhen 518057, China; liangrui1229@126.com (R.L.); h1094157@connect.hku.hk (M.M.H.Y.); hysngan@hku.hk (H.Y.S.N.); 2Department of Obstetrics & Gynaecology, LKS Faculty of Medicine, The University of Hong Kong, Hong Kong, China; hff1314@hku.hk (F.H.); pljiao@connect.hku.hk (P.J.); kklchan@hku.hk (K.K.L.C.)

**Keywords:** ovarian cancer, peritoneal metastases, BCL2A1, hypoxia, intrinsic cell apoptosis

## Abstract

**Simple Summary:**

Cancer metastasis is still the main cause of cancer-related mortality. Peritoneal metastases are the first presentation of advanced ovarian cancer, and metastatic cancer cells tend to disseminate trans-peritoneally in the peritoneal cavity. Hypoxia is a critical factor in governing transcoelomic metastases of ovarian cancer. Therefore, targeting hypoxia appears to be a promising approach to arrest cancer metastasis. In the present work, we identified that BCL2A1, a BCL2 family member, is significantly induced by hypoxia and other physiological stresses by NF-κB signaling and followed by a gradual degradation. The upregulated BLC2A1 has been shown to enhance ovarian cancer survival, tumor growth, and tumor dissemination by suppressing intrinsic cell apoptosis. These data indicate BCL2A1 is an early response factor in the stressed tumor microenvironment, and targeting BCL2A1 may be a potential therapeutic approach in eradicating peritoneal metastases of ovarian cancer.

**Abstract:**

Emerging evidence indicates that hypoxia plays a critical role in governing the transcoelomic metastasis of ovarian cancer. Hence, targeting hypoxia may be a promising approach to prevent the metastasis of ovarian cancer. Here, we report that BCL2A1, a BCL2 family member, acts as a hypoxia-inducible gene for promoting tumor progression in ovarian cancer peritoneal metastases. We demonstrated that BCL2A1 was induced not only by hypoxia but also other physiological stresses through NF-κB signaling and then was gradually reduced by the ubiquitin-proteasome pathway in ascites-derived ovarian cancer cells. The upregulated BCL2A1 was frequently found in advanced metastatic ovarian cancer cells, suggesting its clinical relevance in ovarian cancer metastatic progression. Functionally, BCL2A1 enhanced the foci formation ability of ovarian cancer cells in a stress-conditioned medium, colony formation in an ex vivo omental tumor model, and tumor dissemination in vivo. Under stress conditions, BCL2A1 accumulated and colocalized with mitochondria to suppress intrinsic cell apoptosis by interacting with the BH3-only subfamily BCL2 members HRK/BAD/BID in ovarian cancer cells. These findings indicate that BCL2A1 is an early response factor that maintains the survival of ovarian cancer cells in the harsh tumor microenvironment.

## 1. Introduction

Ovarian cancer is one of the deadliest gynecological cancers in females worldwide [1]. The high mortality rate of ovarian cancer is attributed to the lack of reliable biomarkers and the subsequent late detection of the disease [1]. In contrast to other solid tumors, advanced ovarian cancer is prone to transcoelomic spreading, which cancer cells spread through the surfaces of peritoneal cavities and the omentum [2,3]. Accumulating evidence has suggested that advanced ovarian cancer with peritoneal metastases is associated with the development of chemoresistance and high recurrence, leading to poor prognosis of patients with this disease [4,5]. The majority of cancer-related deaths are due to the metastasis of tumor cells from the initial primary tumor site to distant sites [6]. Hence, the delineation of the molecular mechanisms underlying ovarian cancer peritoneal metastases will help identify therapeutic targets and provide alternative or supplemental therapeutic regimens.

Emerging evidence has suggested that the tumor microenvironment plays a critical role in the malignant transformation and metastatic progression of cancer cells and the cell response to cytotoxic or targeted therapies [7,8]. Hypoxia or low oxygen tension is known as an adverse prognostic factor in a variety of solid tumors and plays a crucial role in the tumor microenvironment (TME) to facilitate metastatic progression [9,10]. Accumulating evidence has suggested that hypoxia is a key factor in the regulation of cell survival, tumor colonization, and invasiveness of ovarian cancer peritoneal metastases [11,12,13]. The response of cancer cells to hypoxia is a significant factor that contributes to their adaptation to hostile microenvironments. Such responses include genetic/epigenetic alterations during tumor progression to increase cell survival capacity as well as other oncogenic behaviors, such as angiogenesis, vasculogenesis, epithelial to mesenchymal transition (EMT), cell motility, invasiveness, chemoresistance, radiotherapy resistance, cell death resistance, and metabolic reprogramming. [3]. The effects of hypoxia on these cell behaviors make hypoxia a promising target for cancer therapy development [14,15].

Here, we show that a BCL2 family member, BCL2A1, acts as a stress-inducible factor that protects ovarian cancer cells against hypoxia-induced cell death by transcriptome profiling analysis. Our findings were verified in human ovarian cancer tissues, where BCL2A1 was associated with advanced and high-grade metastatic ovarian cancer. Our findings in this study showed that BCL2A1, which is induced by NF-κB signaling under various conditions of physiological stress, could enhance the cell proliferation, cell migration/invasion, and tumor growth of ovarian cancer cells using in vitro, ex vivo, and in vivo tumorigenic models. Importantly, the subcellular localization of BCL2A1 to mitochondria further confirmed that BCL2A1 is a crucial regulator of the mitochondria-mediated apoptotic pathway, indicating that it is a potential therapeutic target for the treatment of the peritoneal metastases of ovarian cancer.

## 2. Materials and Methods

### 2.1. Cell Lines and Clinical Samples

Five human ovarian cancer cell lines, OVCA420, OVCA429, OVCA433, SKOV3, and ES-2, and human embryonic kidney 293 (HEK 293) cells were purchased from American Type Culture Collection (ATCC, Manassas, VA, USA). Two ovarian cancer cell lines, A2780s and A2780cp, two cervical cancer cell lines, OV2008 and C13* (obtained from Prof. Benjamin Tsang), two human ovarian surface epithelial (HOSE) cell lines, HOSE-11-12 and HOSE-96-9-18 (obtained from Prof. George Tsao), and HEK293FT cells (obtained from Dr. Cherie Lee). All the cell lines were maintained in Dulbecco’s modified Eagle’s medium (Invitrogen Life Technologies, Carlsbad, CA, USA) supplemented with 10% (*v*/*v*) fetal bovine serum (FBS) (Invitrogen, Gibco, Gaithersburg, MD, USA) and 100 U/mL penicillin/streptomycin (P/S) (Invitrogen Life Technologies, Carlsbad, CA, USA) in an incubator at 37 °C with a humidified atmosphere of 5% CO_2_ and 95% air. All the clinical samples were obtained from the Department of Obstetrics and Gynaecology of The University of Hong Kong at Queen Mary Hospital with the prior approval of the Institutional Review Board of the University of Hong Kong/Hospital Authority Hong Kong West Cluster (HKU/HA HKW IRS) (IRS Reference Number: UW 11-298 and UW 20-256).

### 2.2. Plasmids

pCMV6-BCL2A1-DDK (RC201965, OriGene, Austin, TX, USA) was used for the overexpression of BCL2A1. An shRNA (BCL2A1) lentiviral plasmid targeting endogenous BCL2A1 (sc-37285, Santa Cruz Biotechnology, Dallas, TX, USA) was used to establish BCL2A1-knockdown cells. CRISPR/Cas9-mediated gene knockout of HIF-1α and BCL2A1 plasmids was established by pSpCas9(BB)-2A-GFP (PX458) (Plasmid #48138, Addgene, Watertown, MA, USA) carrying custom sgRNA oligonucleotides synthesized by Integrated DNA Technologies (IDT, Coralville, IA, USA) (please see the sequences of sgRNAs in Appendix A). The NF-κB p65 (RELA)-expressing plasmid pRSV-p65 was obtained from Addgene (Plasmid #106453), the HIF-1α (RC202461)-expressing plasmid was purchased from OriGene (OriGene, Rockville, MD, USA), and RELA siRNA was ordered from Integrated DNA Technologies (IDT).

### 2.3. RNA Extraction and Real-Time Quantitative PCR (qPCR) Analysis

Total RNA was extracted using TRIzol reagent (Invitrogen, Waltham, MA, USA) according to the manufacturer’s instructions. Complementary DNA (cDNA) was synthesized using High-Capacity cDNA Reverse Transcription Kits (Applied Biosystems, Waltham, MA, USA). BCL2A1 expression was evaluated by qPCR in an ABI PRISM 7500 system (Applied Biosystems) using TaqMan gene expression assays. Human BCL2A1 (Assay ID: Hs00187845_m1) and GAPDH (Assay ID: Hs02786624_g1) were used as the internal controls. Target gene expression was quantified through the comparative delta CT method.

### 2.4. XTT Cell Proliferation Assay

The Cell Proliferation Kit II (XTT) (Roche, Basel, Switzerland) was used to measure the cell growth rate. The whole assay was performed following the manufacturer’s instructions. The absorbance reading on the first day was used as a baseline to normalize the cell growth in response to each treatment on the following days.

### 2.5. Soft Agar Assay

In this assay, ovarian cancer cells were grown in full medium containing 0.3% agarose on the top layer and 1% agarose on the bottom layer. The colonies were monitored every three days. The colony number was counted, and photos were taken under a microscope.

### 2.6. Anoikis Assay

The anoikis assay mimics cell death that is induced by deprivation of the extracellular matrix. A total of 1 × 10^6^ cells per well were seeded in ultra-low attachment 6-well plates pre-coated with poly-Hema. Cells grown in suspension were cultured for 24–48 h before analysis by Western blots or TUNEL assay.

### 2.7. Co-Immunoprecipitation and In Vivo Ubiquitination Assays

A co-immunoprecipitation assay was performed as described previously [16]. Briefly, HEK293 cells were transfected with HA-tagged BCL2A1 and Flag-tagged HRK cDNA vector plasmids. Thirty-six hours after transfection, the cells were harvested and lysed with NP40-RIPA lysis buffer containing proteinase inhibitors. One milligram of protein in 500 μL of cell lysate was precleared with normal IgG for half an hour at 4 °C. Primary antibodies, including anti-HA, anti-Flag, and normal IgG (2 μg) (Sigma-Aldrich, St. Louis, MI, USA), were added to each sample and incubated for 4 h on a shaker at 4 °C. Furthermore, the in vivo ubiquitination assay was also performed as described previously [17]. For details, please see Appendix A.

### 2.8. Luciferase Reporter Assays

An NF-κB luciferase reporter assay was conducted to determine the activity of the NF-κB signaling pathway after treatment with hypoxia (0.5% O_2_) in an Incubator Subchamber (BioSpherix Ltd., Parish, NY, USA). To study the response of the NF-κB signaling activity after hypoxia stimulation (0.5% O_2_), 2 × 10^4^ HEK293 or 1 × 10^4^ OVCA433 cells were seeded into 24-well plates 24 h before Lipofectamine™ 3000^®^ (Invitrogen)-mediated transfection. A mixture of 100 ng NF-κB luciferase reporter plasmid DNA, 5 ng of Renilla plasmid DNA (Promega, Madison, WI, USA), and Lipofectamine™ 3000^®^ was added dropwise to the cells. After 36 h of transfection, the cells were treated with hypoxia (0.5% O_2_) for 4 h with or without the addition of the NF-κB inhibitor Bay 11-7082 (10 μM). The luciferase activity was detected using the Dual-Luciferase^®^ Reporter Assay (Promega, Madison, WI, USA). Similarly, the BCL2A1 promoter-luciferase reporter assay was performed to detect the transcriptional activity of HIF-1α and NF-ĸB by using the BCL2A1-promoter plasmid with Gaussia luciferase (GLuc) purchased from GeneCopoeia (Product ID: HPRM34512, GeneCopoeia, Rockville, MD, USA).

### 2.9. In Situ Proximity Ligation Assay

OVCA433 cells in 6-well plates were co-transfected with Flag-tagged BCL2A1 plasmid (pAcGFP1-Mito plasmid, Origene, Rockville, MD, USA) and mitochondria labeling plasmid 24 h before being seeded into 8-well chamber slides. Twenty-four hours later, the cells were washed once with PBS and then fixed with 2% PFA, followed by permeabilization using 0.02% Triton X-100 in PBS. Primary antibodies against Flag-M2 (Santa Cruz Biotechnology, Dallas, TX, USA), BCL2A1 (ab45413), BAD (ab62465), BID (ab272880), HRK (ab45419) (Abcam, Cambridge, MA, USA) were diluted at a ratio of 1:200 in 1% BSA. For each well, both diluents of Flag-M2 (Sigma-Aldrich, St. Louis, MI, USA) and one of the other antibodies were added. Flag-M2 diluent alone was used as the negative control. Then, 1× PLA probe solution (Sigma-Aldrich, St. Louis, MI, USA) was added to each well, and the slides were incubated in a preheated humidity chamber for 1 h at 37 °C. After ligation and amplification, the cells were further stained with DAPI and mounted for further confocal analysis (40×). The positive protein interaction signals appeared as red dots, and the mitochondria appeared as green GFP signals. Finally, the slides were visualized using confocal microscopy with amplification at 40×. The red dots indicate that the two targeted proteins interact with each other, and the green dots represent the mitochondria. The results were analyzed by ZEN software (Zeiss, Germany).

### 2.10. Western Blot Analysis

Denatured protein lysates were separated by sodium dodecyl sulfate-polyacrylamide gel electrophoresis (SDS-PAGE), followed by electrophoretic transfer to polyvinylidene difluoride (PVDF) membranes using a Mini Trans-Blot Electrophoretic Transfer Cell (BioRad, Hercules, CA, USA). The PVDF membranes were then blocked with 5% (*w*/*v*) skim milk in 1× TBS-T solution (Tris-buffered solution (TBS) + 0.1% Tween 20) and incubated with primary antibodies against BCL2A1 (ab45413) and HIF-1α (ab82832) (Abcam, Cambridge, MA, USA), hydroxyl-HIF-1α (#3434), NF-κB p65 (#8242), IκB α (#4814), p-IκB α (#2859), Cytochrome c (#11940), cleaved-PARP (#5625) and cleaved-Caspase 3 (#9664) (Cell Signaling Technology, Danvers, MA, USA), and β-actin (Sigma-Aldrich, St. Louis, MI, USA) in 1% BSA in TBS-T overnight at 4 °C. The membranes were then incubated with mouse or rabbit polyclonal secondary antibodies (Amersham Pharmacia Biotechnology, Buckinghamshire, UK) in 1% skim milk in TBS-T for 1 h at room temperature. The targeted protein signals were detected by incubation with ECLTM Western Blotting Detection Reagents (Bio-Rad, Hercules, CA, USA) and developed on X-ray film (FUJIFILM Medical Systems, Minato City, Tokyo, Japan). The intensity of Western blot bands and respective ratios were quantified using ImageJ (1.53j) (https://imagej.net/, accessed on 13 July 2021).

### 2.11. Immunofluorescence Staining

Cells grown in 8-well chamber slides were washed with ice-cold PBS once and then fixed with 2% PFA at room temperature for 20 min. After permeabilization with 0.1% Triton X-100 in PBS, the cells were blocked with 2% BSA in TBS for 1 h at room temperature. After blocking, the cells were incubated with primary antibodies diluted in 1% BSA overnight at 4 °C and then washed 3 times for 5 min each at room temperature. Secondary antibody (Life Technologies) incubation was performed in the dark at room temperature for 1 h. Finally, the cells were stained with DAPI and processed for confocal analysis (40×). The fluorescence intensity was analyzed by ZEN software (Zeiss, Germany).

### 2.12. Immunohistochemical Staining

A commercial ovarian cancer tissue array (OV481, US Biomax, Derwood, MD, USA), which includes 12 samples of ovarian tumors of various grades paired with 12 unmatched healthy adjacent tissues, was used for clinical analysis of BCL2A1 expression. The tissue array in paraffin was first treated with deparaffinization using xylene. A gradient of alcohol and distilled water was used for rehydration. Antigen retrieval was performed by soaking in antigen retrieval buffer (10 mM sodium citrate buffer, pH 6.0, 1 mM EDTA, pH 8.0) at 100 °C for 15 min, and the sections were then incubated in TBS containing 10% hydrogen peroxidase (H_2_O_2_) for 20 min. The sections were then blocked with normal IgG (30 min, room temperature), BCL2A1 (ab45413, Abcam, Cambridge, MA, USA) (4 °C overnight), and secondary antibody (30 min, room temperature). After each antibody incubation period, the sections were washed three times with TBST for 10 min. Finally, the sections were counterstained with hematoxylin and eosin and then visualized with a DAKO EnVision^®^ system (DAKO, Carpinteria, CA, USA). An Aperio ScanScope CS System was used to scan and create digital slides for further analysis.

### 2.13. Ex Vivo Colonization Assay

Omenta tissues resected from SCID mice were allowed to adhere to 12-well plates precoated with Cell-Tak Adhesive. Ovarian cancer cells labeled with GFP were added to the omental culture medium at a density of 5 × 10^5^ cells/3 mL. The culture was incubated at 37 °C with a humidified atmosphere of 5% CO_2_ and 95% air for 24 h, followed by washing using PBS to remove the nonattached ovarian cancer cells, which were then replated in a new 12-well plate and cultured in DMEM/F12 with 20% FBS for 2 weeks. The medium was changed every 4 days, and the number of colonies was counted under a ZOE^TM^ Fluorescence Cell Imager (Bio-Rad).

### 2.14. In Vivo Tumor Dissemination Mouse Model

Ovarian cancer OVCA433 or ES-2 cells with CRISPR-mediated BCL2A1 knockout (BCL2A1^−/−^) and scrambled control cells (1 × 10^6^ cells/200 µL) were implanted intraperitoneally (*i.p*.) into four-week-old female SCID mice as described previously [18]. The mice were sacrificed four weeks after post-injection, and the tumor nodules were counted by stereomicroscope. The omentum and mesenterium were harvested, weighed, and fixed before being embedded in paraffin blocks. The animal study was performed according to the guidelines approved by The Committee on the Use of Live Animals in Teaching and Research of The University of Hong Kong (CULATR number: 3655-15).

### 2.15. Statistical Analysis

The statistical analyses in this study were performed using GraphPad Prism 5.0 (San Diego, CA, USA). The data presented in this study were analyzed by unpaired *t*-test, one/two-way analysis of variance (ANOVA), and Tukey post hoc comparison. All the data were collected from at least three independent experiments. Each was performed in triplicate, except otherwise mentioned. The data are expressed as the mean ± S.E.M., and *p* < 0.05 was considered statistically significant.

## 3. Results

### 3.1. BCL2A1 Is an Early Response Gene to Hypoxia in Ovarian Cancer Cells

To identify genes and pathways that are altered in gynecological cancer cells after hypoxic stimulation, transcriptional profiling (GeneChipTM Affymetrix Human Genome U133 Plus 2.0 Array) was performed on a pool of gynecological cancer cell lines (OVCA433, A2780cp, SKOV3, OV2008, and C13* cells) treated with hypoxia (0.5% O_2_, 5% CO_2_, 24 h) or normoxia, which was used as a control. Gene expression was shown as a log-fold-change cutoff ±2-fold based on normalization with hypoxia vs. normoxia (Appendix A). We have deposited the microarray data into ArrayExpress (accession E-MTAB-9730). After gene profile analysis, 26 upregulated genes and 22 downregulated genes with a minimum fold change difference of 2 were selected (Figure 1A). The Gene Ontology Consortium indicated that these differentially expressed genes (DEGs) were associated with four signaling pathways, including the WNT signaling pathway, inflammation mediated by chemokine and cytokine signaling pathways, the apoptosis signaling pathway, and heterotrimeric G-protein (Gi α- and Gs α-mediated) signaling pathways (Appendix A). Given that apoptotic resistance is one of the essential hallmarks in cancer metastasis, the apoptosis signaling pathway was selected for further study. Among apoptosis pathway components, BCL2A1, one of the hypoxia-responsive genes of the BCL2 family, was found to be significantly induced by 5.5-fold in hypoxia-treated gynecological cancer cells (Figure 1A and Appendix A). However, no other BCL2 family members showed a significant fold change in expression (>2-fold) after hypoxia treatment of ovarian cancer cells.

Domcke et al. recently reported that both A2780cp and SKOV3 cells are ovarian adenocarcinoma cell lines instead of high-grade serous ovarian cancer (HGSOC) cell lines [19]. To confirm the expression patterns of BCL2A1 in different ovarian cancer cells after hypoxia treatment (0.5% O_2_), we performed qPCR analysis on OVCA433, ES-2, and A2780cp cells. The results showed that the mRNA levels of BCL2A1 were induced by 3.5-fold in OVCA433 cells, 3.9-fold in ES-2 cells, and 4.3-fold in A2780cp cells after 4 h of hypoxic treatment, and these results were immediately followed by a gradual reduction in the BCL2A1 levels in all cell lines (Figure 1B). Although the BCL2A1 protein induction was slightly delayed, Western blot analysis consistently revealed an upregulated expression of BCL2A1 at 4 h of hypoxia treatment, and this pattern was prolonged for 12 h before reduction to the basal level at 24 h of hypoxia treatment (Figure 1C). We also noted that the induction of BCL2A1 expression coincided with the induction of HIF-1α at 2 h after hypoxia treatment (Figure 1C), indicating that BCL2A1 may be an early response factor of hypoxia in ovarian cancer cells. It is known that HIF-1α and its target genes compose the most important signaling network in response to hypoxia [20,21]; however, it is unknown whether the induction of BCL2A1 by hypoxia is induced through HIF-1α signaling. JASPAR transcription factor analysis (http://jaspar.genereg.net/, accessed on 13 July 2021) using HIF-1α and the promoter sequence of BCL2A1 showed that there was no single binding site of HIF-1α in the BCL2A1 promoter region. To prove this, we performed a BCL2A1 promoter-luciferase assay with transient transfection of the HIF-1α-expressing plasmid in OVCA433 cells. We found that elevation of HIF-1α did not upregulate the level of BCL2A1 mRNA (Figure 1D). Furthermore, using CRISPR/Cas9-mediated HIF-1α gene knockout and co-treatment of CoCl_2_-simulated hypoxia in ovarian cancer cells, OVCA433 and ES-2, the changes of HIF-1α also did not alter the protein expression level of BCL2A1 (Figure 1E). These results showed that HIF-1α signaling is not involved in the induction of BCL2A1 by hypoxia.

### 3.2. NF-κB Transcriptionally Regulates Hypoxia-Induced BCL2A1 Upregulation

Because NF-κB is also an early response signal after infection and stress [20,22], it is of interest to investigate whether the induction of BCL2A1 is mediated by NF-κB signaling after hypoxia treatment. JASPAR transcription factor analysis revealed that four putative binding sites of NF-κB p65 (RELA) were localized in the promoter region of BCL2A1 (Appendix A). To confirm this, a luciferase reporter vector containing the luciferase gene under the control of the BCL2A1 promoter was transiently co-transfected with a RELA-expressing vector (0, 50, and 100 ng) into OVCA433 cells. The results showed that BCL2A1 promoter activity was enhanced with the increased expression of RELA in a dose-dependent manner (*p* < 0.01) (Figure 2A). Additionally, increased expression of RELA significantly elevated the expression of BCL2A1 (Figure 2B), whereas the depletion of RELA using siRNAs targeting RELA prevented BCL2A1 induction by hypoxia in OVCA433 cells (Figure 2B). To understand the relationship between BCL2A1 and NF-κB p65 (RELA), Western blot analysis firstly revealed the expression patterns of BCL2A1 and NF-κB p65 (RELA) in a panel of ovarian cancer cell lines and two HOSE cell lines (Figure 2C). IκB kinases are the key negative regulators of NF-κB pathway and their phosphorylation status are usually used as an indicator of the NF-κB signaling activity induced by stress stimuli [23]. Intriguingly, the level of phosphorylated IκB (p-IκB) was increased within 2 to 8 h upon hypoxia treatment, followed by gradual reduction (Figure 2D). This indicated that an increase in NF-κB activity initially occurred because of stress induction, whereas it was gradually attenuated due to the loss of stress stimuli. To test this hypothesis, an NF-κB luciferase assay was performed, and the results showed that NF-κB luciferase activity was significantly induced by approximately 2.5-fold in OVCA433 cells (*p * < 0.0001) and 2-fold in A2780cp cells (*p* < 0.0001) after 4 h of hypoxia treatment (Figure 2E). In contrast, in the presence of an NF-κB inhibitor, Bay 11-7082 (10 μM), the relative luciferase activity of NF-κB failed to be stimulated by hypoxia at the same time point (Figure 2E). Consistently, co-treatment of OVCA433 cells with Bay 11-7082 (10 μM) reduced the induction of *BCL2A1* mRNA compared with the control (Figure 2F). Taken together, these results confirmed that the upregulation of BCL2A1 by hypoxia is mediated by the NF-κB signaling pathway.

### 3.3. BCL2A1 Can Be Induced in Ovarian Cancer Cells by Various Physiological Stressors

In the peritoneal metastases of ovarian cancer, metastatic ovarian cancer cells have to overcome various physiological stresses other than hypoxia [24]. To verify whether other physiological stresses can induce BCL2A1, we subjected cultured OVCA433, A2780cp, and SKOV3 ovarian cancer cell lines to serum starvation, ultra-low attachment plates (anoikis assay), and glucose-free media. Western blot analysis showed that BCL2A1 expression could be induced during 4–16 h of serum starvation, with the highest induction at 8 h in both OVCA433 and A2780cp cells when cultured in serum starvation media (0%FBS) (Figure 3A). Using an anoikis assay, BCL2A1 was induced at 24 h in OVCA433 and A2780cp cells (Figure 3B). However, BCL2A1 was induced at a much higher level at 48 h in OVCA433 cells (Figure 3B). In comparison, there was a reduction in the BCL2A1 protein level at 48 h in A2780cp cells (Figure 3B), indicating differential BCL2A1 auto-regulation in different cell contexts. Similarly, BCL2A1 was induced at 0.5 h and reached its highest level at 1 h, followed by a gradual reduction in OVCA433 cells when cultured in glucose-free medium (Figure 3C). On the other hand, BCL2A1 was induced at 2 h and reached its highest level at 5 h before exhibiting a moderate reduction in its protein level in SKOV3 cells when cultured in the same glucose-free medium (Figure 3C). Based on the discrepancy in the mRNA and protein levels of BCL2A1 in ovarian cancer cells after various stressful challenges, it was hypothesized that the induction of the BCL2A1 protein by stressors might also involve posttranscriptional modifications, such as ubiquitin-proteasome degradation. To validate this hypothesis, we tested whether co-treatment with the proteasome inhibitor MG132 could suppress protein degradation in hypoxia-treated cells. Indeed, Western blot analysis showed that MG132 inhibited the reduction in the BCL2A1 protein levels in OVCA433 cells after hypoxic treatment for 0–16 h (Appendix A). Further investigation using an in vivo ubiquitination assay revealed that hypoxia treatment inhibited the ubiquitin level of the BCL2A1 protein at 16 h (Figure 3D), confirming that ubiquitin-proteasome degradation is involved in modulating the BCL2A1 protein levels in addition to transcriptional regulation.

### 3.4. BCL2A1 Is Frequently Upregulated in Advanced Metastatic Ovarian Cancers

Previous studies reported that there was an association of high BCL2A1 expression with metastatic melanoma and hepatocellular carcinoma [25,26], which prompted us to consider the clinical function of BCL2A1 in ovarian cancer. By analyzing The Cancer Genome Atlas (TCGA) datasets, we found a 4-fold increase in the BCL2A1 mRNA levels in ovarian cancer samples compared with normal ovary samples (Appendix A). To further investigate whether upregulated BCL2A1 is associated with metastatic ovarian cancers, we first examined the expression of BCL2A1 in clinical ovarian cancer tissues or cells obtained from the primary ovaries, ascites, and lavage of ovarian cancer patients by Western blot analysis. The results showed that significantly higher expression of BCL2A1 was observed in ovarian cancer cells isolated from lavage or ascites than in those isolated from primary ovarian tumors (Figure 3E). By quantifying the band intensities of BCL2A1, we noted a 2.7-fold increase in BCL2A1 in metastatic ovarian cancer cells isolated from lavage/ascites compared with those isolated from primary ovarian tumors (Figure 3F). In two pairs of patient samples (P15 and P21), the analysis indicated that the relative expression of BCL2A1 after normalization by β-actin was 1.6–3.7-fold higher in metastatic ovarian cancer cells isolated from lavage/ascites than in those from the paired primary ovarian tumors (Figure 3G). Immunohistochemical (IHC) analysis using a commercial ovarian cancer tissue array (OV481) was subsequently performed to confirm the BCL2A1 expression pattern in clinical ovarian cancer specimens. The results showed that a higher expression level of BCL2A1 was observed in all cancerous tissues compared with normal tissues and was positively correlated with high-grade and advanced ovarian cancers compared with the level of BCL2A1 in low-grade and early-stage ovarian cancers (Figure 3H). These findings suggest that a higher level of BCL2A1 is required for metastatic ovarian cancer cells associated with high-grade and advanced ovarian cancers.

### 3.5. BCL2A1 Is Required for the Enhanced Cell Viability of Ovarian Cancer Cells under Stress

To further study the functional roles of BCL2A1 in ovarian cancer progression, BCL2A1 stable cell lines with overexpression or gene knockdown were established by lentiviral expression of BCL2A1 cDNA or shRNAs (Figure 4A,B). To investigate whether high BCL2A1 expression was required for ovarian cancer progression, the effect of BCL2A1 on cell viability under different types of stress, such as hypoxia and serum starvation, was examined by XTT cell proliferation assay. The results showed that the relative cell growth rate of OVCA433 cells and A2780cp cells with overexpression of BCL2A1 was 1.43-fold (*p* = 0.002) and 1.25-fold (*p* < 0.0001) higher than that of the corresponding control OVCA433 and A2780cp cells, respectively, after hypoxia treatment (Figure 4C). In contrast, knockdown of BCL2A1 by the shRNA approach with the shBCL2A1-1 and shBCL2A1-2 clones reduced the viability of OVCA433 cells to 60–68% (*p* < 0.001), and use of the shBCL2A1-1 and shBCL2A1-2 clones reduced the viability of A2780cp cells to 50–57% (*p* < 0.01) compared to their scrambled controls (Figure 4C). This indicates that BCL2A1 enhances the viability of ovarian cancer cells under hypoxic conditions. Given that ovarian cancer cells also suffer from growth factor deprivation, the effect of BCL2A1 on cell growth under low serum culture conditions (0.5% FBS) was also determined. XTT cell proliferation assays showed that BCL2A1 increased cell viability by 1.36-fold (*p* = 0.004) in OVCA433 cells and 1.50-fold (*p* = 0.003) in A2780cp cells in low-serum culture medium (Figure 4D). In contrast, depletion of BCL2A1 by the shRNA approach reduced cell viability by 43–64% (*p* < 0.001) in OVCA433 cells and 55–60% (*p* = 0.02 and *p* = 0.004) in A2780cp cells compared to scrambled control cells (Figure 4D). To assess the effect of BCL2A1 on cell viability under long-term stress, a foci formation assay was performed in ovarian cancer cells with BCL2A1 overexpression or knockdown. The results showed that the foci number of BCL2A1-expressing cells was increased by 1.4-fold (*p* = 0.01) in OVCA433 cells and 1.6-fold (*p* = 0.01) in A2780cp cells compared with that of the vector control cells, confirming that BCL2A1 enhanced the foci formation capacity of ovarian cancer cells (Figure 4E). Consistently, the depletion of BCL2A1 by shRNA reduced the foci number by 45–50% (*p* < 0.01) in OVCA433 cells and 55–60% (*p* < 0.001) in A2780cp cells (Figure 4F). These results suggest that BCL2A1 plays a role in cancer cell aggressiveness by increasing cell survival and growth under stress conditions. On the other hand, it is of interest to examine BCL2A1 in ovarian cancer cells in a non-stressed culture. To this end, XTT cell viability assay demonstrated that ectopic expression of BCL2A1 reduced the proliferation rate by 21% in OVCA433 cells (*p* < 0.05) and 22% in A2780cp cells (*p* < 0.05) (Figure 4G). In contrast, knockdown of BCL2A1 by shRNAs showed a significant increase in the cell proliferation rate of two folds in sh-BCL2A1-1 cells and sh-BCL2A1-2 cells (*p* < 0.05) (Figure 4G). Cell cycle analysis revealed that overexpression of BCL2A1 led to a 40% increase in G1 phase of cell cycle, whereas the depletion of BCL2A1 by shRNAs caused 20% reduction of cell population in G1 phase but 25% in S phase of cell cycle in OVCA433 cells from a non-stressed culture (Appendix A). These results suggest that increased BCL2A1 expression leads to a G1 phase cell cycle arrest in ovarian cancer cells under non-stressed culture conditions. To support this notion, Western blot analysis showed that the expressions of p18 (INK4c) and p27 (Kip1) negative cell cycle regulators were apparently upregulated by BCL2A1 stably overexpressed in OVCA433 cells (Figure 4H). Contrarily, the depletion of BCL2A1 by shRNAs significantly decreased expression levels of p18(INK4c) and p27(Kip1) in both sh-BCL2A1-1 cells and sh-BCL2A1-2 cells as compared with scrambled control (Figure 4H). These results indicate a G1 phase cell cycle arrest caused by BCL2A1 through the regulation of p18 and p27 in ovarian cancer cells cultured in a non-stressed condition.

### 3.6. BCL2A1 Inhibits Cell Apoptosis and Enhances the Oncogenic Properties of Ovarian Cancer Cells

According to previous reports on BCL2A1 functions in different cancer cells or normal cells [27,28], it was hypothesized that BCL2A1 could augment the capability to resist stress-induced cell apoptosis. To test this notion, a TUNEL assay showed that overexpression of BCL2A1 significantly decreased the number of apoptotic OVCA433-BCL2A1 cells to 21%, 11%, and 9% under hypoxia, glucose-deprivation, and anoikis conditions, respectively (*p* < 0.01) (Figure 5A). Conversely, knockdown of BCL2A1 in A2780cp cells elevated the apoptotic rate by 10–11 folds (*p* < 0.01) under both hypoxia and glucose-deprivation starvation conditions and by approximately 2.5 folds (*p* < 0.01) in the anoikis assay (Figure 5B).

Although metastatic ovarian cancer cells can spread throughout the peritoneal cavity, the omentum is the preferred site for ovarian cancer metastatic colonization [29,30]. We thus performed a soft agar assay to test the anchorage-independent colony formation ability of BCL2A1 in ovarian cancer cells. The results showed that overexpression of BCL2A1 significantly increased the size and colony numbers of OVCA433 and A2780cp cells by 3- and 2.5-fold, respectively (*p* < 0.001) (Figure 5C). In contrast, knockdown of BCL2A1 markedly reduced the size and number of colonies in the A2780cp cells with stable BCL2A1 knockdown by 50% and 40%, respectively (*p* < 0.0001) (Figure 5D). These results suggest that BCL2A1 plays a role in cancer cell aggressiveness by increasing cell survival and growth in environments without proper anchorage.

Moreover, to study the role of BCL2A1 in omental metastases of ovarian cancer cells, an ex vivo omental tumor-seeding model was used. In this model, GFP-labeled cancer cells were allowed to grow and form colonies on a piece of excised murine omentum obtained from SCID mice. At the endpoint, the colony size and number of tumor colonies were observed by a fluorescence stereomicroscope (Nikon, Melville, NY, USA). The results showed that overexpression of BCL2A1 in ES-2 cells increased both the size and number of colonies by 3-fold compared with the vector control (*p* = 0.011) (Figure 5E). In contrast, depletion of BCL2A1 in A2780cp cells reduced the colony number and size by four-fold compared with the scrambled control (*p* = 0.0009) (Figure 5E). Taken together, these results indicate that BCL2A1 could enhance the tumor colonization ability of ovarian cancer cells in omental metastasis.

### 3.7. BCL2A1 Suppresses Intrinsic Apoptosis by Interacting with HRK/BAD/BID

The findings above suggest that BCL2A1 can enhance cell survival in response to various forms of physiological stress. Hence, it is of interest to examine whether BCL2A1 inhibits cell apoptosis, similar to other BCL2 family members, to control cell sensitivity to stress-induced apoptosis [31]. As expected, overexpression of BCL2A1 in ovarian cancer cells markedly prevented cleavage of PARP and Caspase-3 in ovarian cancer cells exposed to serum starvation, hypoxia, or anoikis-mediated stresses (Figure 6A). In contrast, knockdown of BCL2A1 elevated the cleavage of PARP in ovarian cancer cells under various physiological stress conditions (Figure 6B). BCL2 family proteins usually converge at the mitochondrial outer membrane to modulate the intrinsic apoptosis pathway [31]. Consistently, fluorescence confocal microscopy revealed that BCL2A1 colocalized in mitochondria (Appendix A). The co-localization signals between BCL2A1 and mitochondria were further enhanced by ~20% when ovarian cancer cells were co-cultured in low serum (0.5% FBS), glucose-free, and hypoxia-mediated stress conditions for 4 h (Figure 6C). To confirm whether BCL2A1 inhibited the intrinsic cell apoptosis pathway induced by stress, OVCA433 cells in which BCL2A1 was knocked out by the CRISPR/Cas9 system and the corresponding control cells were treated with hypoxia (0.5% O_2_) or normoxia for 24 h. Western blot analysis of mitochondrial fractions showed that hypoxia triggered the release of cytochrome c from the mitochondria to the cytosol, as reflected by decreased cytochrome c in the mitochondria and increased cytochrome c in the cytosol in the scrambled control (Figure 6D). However, ovarian cancer cells in which BCL2A1 was depleted did not retain cytochrome c in mitochondria, and all cytochrome c was released to the cytosol (Figure 6D), supporting the notion that BCL2A1 suppresses intrinsic cell apoptosis under stress conditions. To delineate the mechanism by which BCL2A1 regulates the apoptotic pathway under stress, BCL2A1-interacting proteins were assessed. Previous studies using in silico mutational scanning have shown that BCL2A1 interacts with HRK to form BCL2A1-HRK complexes [32]. Indeed, a co-immunoprecipitation assay was performed using DDK-tagged HRK (DDK-HRK)- and HA-tagged BCL2A1 (HA-BCL2A1)-expressing plasmids transiently co-transfected into OVCA433 cells. Immunoblotting analysis confirmed the interaction between BCL2A1 and HRK (Figure 6E).

To further delineate the mechanism by which BCL2A1 inhibited apoptosis, an in situ PLA was performed to identify any putative BCL2 family members that interacted with BCL2A1 in mitochondria, which were labeled with the pAcGFP1-Mito marker (green). The results showed that PLA signals (red) were observed for endogenous HRK/BAD/BID (Figure 6F) but not BAX/BAK/PUMA/BIM when interacting with BCL2A1-DDK in mitochondria (Figure 6F). These findings suggest that BCL2A1 suppresses intrinsic cell apoptosis under stress conditions by interacting with the BH3-only protein of activator BID to inactivate BAK/BAX in mitochondrial outer membrane permeabilization (MOMP) [33] and two pro-apoptotic proteins, HRK/BAD [34], in the mitochondria of ovarian cancer cells.

### 3.8. BCL2A1 Is Required for the In Vivo Tumor Dissemination of Ovarian Cancer Cells

To further examine the functional role of BCL2A1 in ovarian cancer cells growing in a complex tumor microenvironment, an in vivo peritoneal ovarian tumor dissemination mouse model was employed. A highly tumorigenic high-grade serous ovarian cancer cell line, OVCA433 cells, was selected for depleting BCL2A1 by the CRISPR/Cas9 knockout system. The scrambled control and BCL2A1^−/−^ OVCA433 cells were then intraperitoneally (i.p.) injected into four-week-old female SCID mice in groups of five. Four weeks after post-injection, all the mice were sacrificed and examined by stereomicroscope. The results showed that tumor nodules were found throughout the peritoneal cavity and were mainly localized in the mesenterium and omentum in all scrambled control groups (Figure 7A). In contrast, only 25% of the mice injected with BCL2A1^−/−^ OVCA433 cells exhibited tumor dissemination in the peritoneal cavity (Figure 7A). The average number of tumor nodules formed by the BCL2A1^−/−^ OVCA433 cells was 8% of that formed by the scrambled control cells (*p* = 0.0002) (Figure 7B). No tumor nodules were formed in the lower left abdominal quadrant of the mouse (injection site). The weight of the tumor nodule formed by BCL2A1^−/−^ OVCA433 cells was 55% less than that formed by the scrambled control cells (*p* = 0.0281) (Figure 7C). The tumor nodules on the omentum and mesenterium were examined by H&E staining, IHC, and Western blot analysis to confirm the expression of BCL2A1 in the tumor cells with or without BCL2A1 gene knockout (Figure 7D,E). Hence, these findings indicate that BCL2A1 is significantly involved in tumor dissemination in the murine peritoneal cavity.

In parallel, a highly metastatic ovarian cancer cell line, ES-2 cells [35], was used for the in vivo tumor dissemination assay. We found that knockout of BCL2A1 in ES-2 (BCL2A1^−/−^ ES-2) cells significantly reduced tumor nodule formation (Figure 7F–H) and improved the survival rate of SCID mice that were i.p. injected with ES-2 scrambled control cells (Figure 7I). To investigate the relationship between BCL2A1 and patient survival, data from The Cancer Genome Atlas (TCGA) (serous ovarian cancer) were used for Kaplan–Meier survival analysis (Appendix A). The survival analysis showed that patients with serous ovarian cancer with increased BCL2A1 copy numbers had a lower survival probability with a hazard ratio of 1.672 and a log-rank test *p*-value of 0.10 (Appendix A). This result confirmed that the upregulation of BCL2A1 is associated with lower survival ability in ovarian cancer patients.

## 4. Discussion

Hypoxia is one of the fundamental characteristics of solid tumors including ovarian cancer and metastatic colonization usually involves peritoneal metastases of ovarian cancer [36,37]. Thus, targeting hypoxia in cancer therapy has become a promising therapeutic approach to impede peritoneal metastases of ovarian cancer [14,15]. In this study, we identified a BCL2 family member, BCL2A1, as an inducible gene through genome-wide transcriptome profiling using the comparison of ovarian cancer cells treated with hypoxia vis-à-vis normoxia. Notably, prior to a gradual decline of the BCL2A1 expression, endogenous BCL2A1 level was rapidly induced not only by hypoxia but also by other physiological stresses through NF-κB signaling. The prompt initiation and progressive reduction of BCL2A1 expression by various biological stressors indicated an auto-regulation system of BCL2A1 expression in ovarian cancer cells. Prolonged high expression of BCL2A1 reduced the cell proliferation rate by causing G1 arrested in ovarian cancer cells under a non-stressed culture condition. Indeed, the BCL2A1 protein level was regulated by the ubiquitin-proteasome pathway. Functional studies revealed that BCL2A1 enhanced foci formation ability in serum starvation medium, anoikis resistance, tumor colony formation capacity in soft agar, ex vivo cultured mouse omenta, and in vivo tumor dissemination, implying that BCL2A1 was involved in cell survival and other oncogenic properties of ovarian cancer cells, especially under stress conditions. Cytochrome c detection by Western blot analysis and in situ PLA indicated that this process is mediated by effects on the intrinsic apoptotic pathway, where BCL2A1 interacted with and inhibited the BH3-only proteins BID/BAD/HRK. To the best of our knowledge, this is the first report to identify BCL2A1 as a stress-inducible gene in ovarian cancer cells.

The physiological microenvironment in peritoneal metastases of ovarian cancer is very complicated [37]. Inadequate oxygen and nutrients, extracellular matrix abnormalities, and cell detachment reduce the survival of ovarian cancer cells and tumor nodules disseminated throughout the peritoneal cavity [37]. In this study, our findings demonstrated that the expression of BCL2A1 was induced by hypoxia, anoikis, serum starvation, and glucose deprivation, indicating that BCL2A1 was a common stress-inducible factor for the survival of ovarian cancer cells. Previous studies on hypoxia and other stresses showed similar effects on tumor progression, supporting the paradigm that stress drives tumor progression [38]. Consistently, BCL2A1 showed higher expression levels not only in advanced and metastatic ovarian tumors but also in metastatic ovarian cancer cells in ascites/lavage clinical samples. Previous studies on ovarian cancer have shown that metastatic cancer cells encounter stresses such as hypoxia [39] and extracellular matrix detachment [18,40]. The findings reported herein clearly verify that metastatic ovarian cancer cells require higher BCL2A1 expression, facilitating tumor survival and progression in harsh tumor microenvironments (TMEs) with multiple stressful challenges [37].

NF-κB signaling and HIF-1α signaling are two pathways that modulate a number of early response genes upon stresses from the TME [20,37]. Our findings showed that NF-ĸB p65 (RELA) but not HIF-1α had four putative binding sites on the promoter of BCL2A1, indicating that NF-κB signaling was the upstream inducer of BCL2A1 in ovarian cancer cells. Indeed, HIF-1α had no obvious effect on BCL2A1 expression in ovarian cancer cells under hypoxic conditions, suggesting that BCL2A1 induction by hypoxia occurs in a HIF-1α-independent manner. In contrast, it is well known that NF-κB is a sensitive inducible transcription factor in cells [41]. BCL2A1 was induced by NF-κB signaling after TNF-α treatment in cervical cancer cells [42]. Similarly, herein, a BCL2A1 promoter-luciferase assay, overexpression, depletion of RELA by RNAi, or inhibition by pharmaceutical approaches in ovarian cancer cells suggest that NF-ĸB signaling is the upstream regulator of BCL2A1 transcriptional activity. Transcription factors in the NF-κB family are well known to play a crucial role in fighting infections, inflammation, and environmental stresses, controlling developmental processes, cell growth, and apoptosis in normal tissue and cells [43]. This protein family is often constitutively active in multiple human cancers as well as some inflammation-related diseases such as arthritis, asthma, and neurodegenerative diseases [44,45]. Especially in malignant diseases, NF-κB pathway-mediated inflammation has long been associated with both carcinogenesis and progression of cancers, including bladder cancer, cervical cancer, and ovarian cancer [46]. However, the responsive genes of these transcription factors differ greatly depending on different cell types and stresses [47]. To the best of our knowledge, there is no report showing that NF-κB activates the expression of BCL2A1 transcriptionally in ovarian cancer. Therefore, our present study is the first report showing that NF-κB p65 is a transcription factor of BCL2A1 that is responsible for the increased expression of BCL2A1 under various types of TME stresses.

Previous studies on other BCL2 family proteins have found that BCL2 confers an oncogenic ability to cancer cells by increasing cell apoptosis resistance [48,49,50,51]. However, the functional roles of BCL2A1 remain unclear, according to limited reports. As an anti-apoptotic BCL2-like protein, BCL2A1 is one of the critical regulators of the intrinsic cell death pathway, also called the mitochondria-mediated apoptotic pathway [52]. Under apoptotic stress conditions, pro-apoptotic BAX and BAK translocate to the outer membrane of mitochondria and increase outer membrane permeabilization, which initiates intrinsic apoptotic signaling [53]. In this study, subcellular localization experiments indicated that BCL2A1 is mainly a cytosolic protein. When subjected to various types of tumor developments and progression stresses, the BCL2A1 mitochondrial localization was increased. As a pro-survival member of the BCL2 family, BCL2A1 shares similar functions with all its counterparts: sequestering the pro-apoptotic BCL2 proteins and inhibiting their activation. BAX and BAD are effectors of the mitochondrial apoptotic pathway, which forms pores in the outer membrane of mitochondria and allows the release of cytochrome c [54]. This study showed that BCL2A1 interacts with neither BAX nor BAK in ovarian cancer cells, consistent with other studies [51,55]. However, the in situ PLA confirmed that ectopically expressed BCL2A1 interacts with and sequesters activator and sensitizer BH3 proteins, such as BID and BAD/HRK, respectively. BID is an activator of the mitochondrial apoptosis pathway, which directly interacts with and activates BAX and BAK [56]. The sequestration of BCL2A1 and BID might inactivate BAK/BAX to avoid mitochondrial outer membrane per-meabilization (MOMP), a critical apoptotic event through the release of cytochrome c [33]. However, HRK and BAD are sensitizers of BH3-only proteins and pro-apoptotic proteins [57]. The interaction of BCL2A1 with HRK and BAD might have a suppressive effect on other antiapoptotic BCL2 members, such as BCL2, MCL1, and BCL-xL. [57]. Indeed, our study showed that the depletion of BCL2A1 significantly exacerbated the release of cytochrome c induced by hypoxia, which was concomitant with the effects of BCL2A1 in the suppression of intrinsic apoptotic signaling. However, further investigations are needed to confirm the effects of BCL2A1 in the sequestration of BAD/HRK and BID in inducing MOMP.

## 5. Conclusions

This study suggests that BCL2A1 is a novel stress-inducible gene that can be induced by various forms of physiological stresses through NF-κB signaling in metastatic ovarian cancer cells. The induction of BCL2A1 by physiological stresses promotes cell survival and oncogenic properties by inhibiting the intrinsic apoptotic pathway via sequestering the activator and sensitizer BH3 proteins BID/BAD/HRK and eventually inhibiting the release of cytochrome c from mitochondria to the cytosol. Hence, targeting BCL2A1 is likely a potential therapeutic approach in eradicating peritoneal metastases of ovarian cancer.

## Figures and Tables

**Figure 1 cancers-13-04577-f001:**
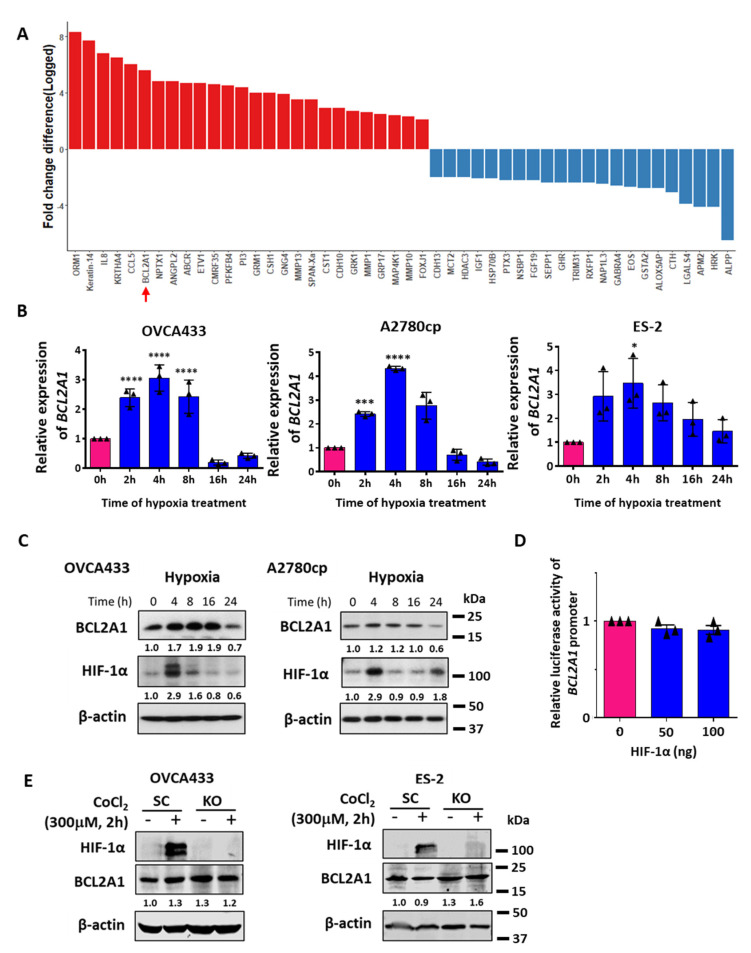
Identification of BCL2A1 as a hypoxia-inducible gene in ovarian cancer. (**A**) A panel of gynecological cancer cell lines (OVCA433, A2780cp, SKOV3, OV2008, and C13*) was treated with hypoxia (0.5% O_2_, 24 h) or normoxia. The gene expression profiling was analyzed using the GeneChipTMAffymetrix Human Genome U133 Plus 2.0 Array (performed by the Center for Genomic Sciences, HKU). Genes with red bar were upregulated genes, while genes with blue bar were downregulated genes with 2 folds in expression. The height of the bar indicated the fold change difference of the respective gene. (**B**) QPCR analysis and (**C**) Western blot analysis showed the relative expression changes of BCL2A1 in OVCA433 and A2780cp cells with hypoxia treatment (0.5% O_2_) for various time points as compared with the normoxia (0 h). The expression of HIF-1α was used as the positive indicator of cell response factor to hypoxia stress. β-actin was used as a loading control for Western blot analysis. (* *p* < 0.05 vs. 0 h, *** *p* = 0.001 vs. 0 h, and **** *p* < 0.0001 vs. 0 h). (**D**) BCL2A1 promoter-luciferase plasmid was transiently transfected with HIF-1α expressing plasmid with various doses (0, 50, and 100 ng) into OVCA433 cells and showed that HIF-1α had no effect on transcriptional activity of BCL2A1. (**E**) CRISPR/Cas9 gene knockout system was used to delete endogenous HIF-1α in OVCA433 and ES-2 cells. SC = scrambled control and KO = HIF-1α^−/−^. CoCl_2_ (300 µM, 2 h) (Sigma-Aldrich, St. Louis, MI, USA), a chemical HIF-1α inducer, was used to simulate the hypoxic effect on ovarian cancer cells. For all functional studies, *n* = 3 technical replicates per sample. Data were represented as mean ± S.E.M.

**Figure 2 cancers-13-04577-f002:**
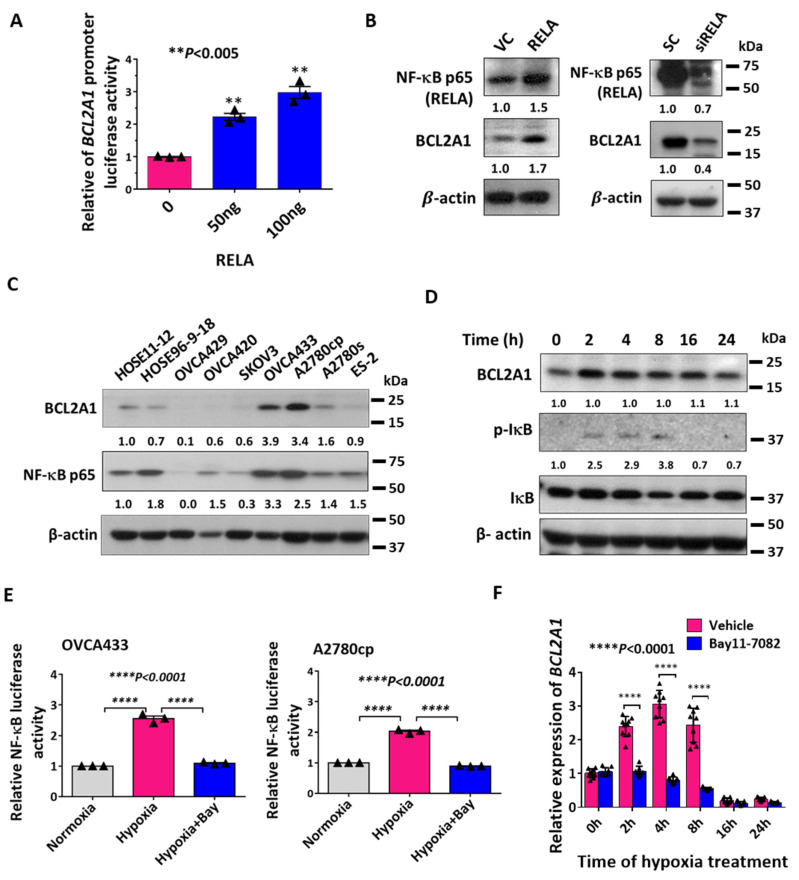
NF-κB signaling is required for the induction of BCL2A1 by hypoxia. (**A**) BCL2A1 promoter-luciferase assay showed that NF-κB p65 (RELA) enhanced the *BCL2A1* transcriptional activity dose-dependently in OVCA433 cells (** *p* < 0.005) as compared with control. (**B**) Western blot analysis showed that ectopic expression of NF-κB p65 (RELA) increased the expression level of BCL2A1, left. Depletion of RELA by siRNAs reduced the expression level of BCL2A1 as compared to the control cells when cultured in hypoxia (0.5% O_2_) for 8 h. (**C**) Western blot analysis displayed that the expression levels of BCL2A1 and NF-κB p65 in a panel of ovarian cancer cells. HOSEs cell lines were used as normal control. (**D**) The induction of BCL2A1 is associated with the activation of NF-κB signaling pathway. OVCA433 cells were treated with hypoxia (0.5% O_2_) for different time points within 0–24 h. (**E**) NF-κB luciferase assay showed that NF-κB transcription activity was increased by hypoxia at 8 h, while NF-κB inhibitor Bay 11-7082 (10 μM) abrogated such NF-κB induction in OVCA433 and A2780cp cells. (**F**) QPCR assay showed that Bay 11-7082 treatment (10 μM) abolished the upregulation of BCL2A1 mRNA level upon hypoxia treatment. For all, *n* = 3 technical replicates per sample. Data were represented as mean ± S.E.M.

**Figure 3 cancers-13-04577-f003:**
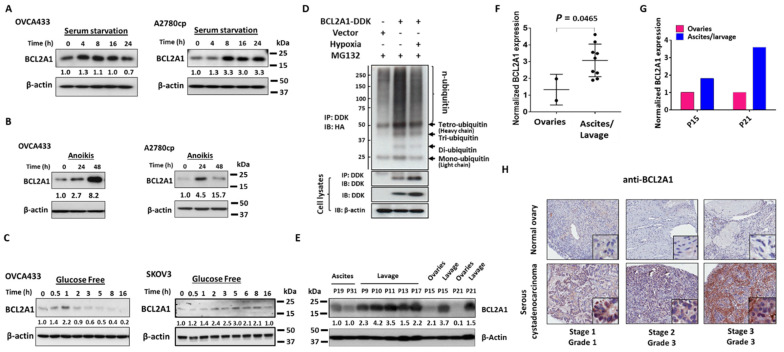
BCL2A1 could be induced by various physiological stresses and associated with advanced metastatic ovarian cancers. (**A**) OVCA433 and A2780cp cells were treated with DMEM containing 0% FBS for different time points (0–24 h). (**B**) OVCA433 and A2780cp cells were cultured in an ultra-low attachment 6-well plate for 24 h and 48 h. (**C**) OVCA433 and SKOV3 cells were treated with glucose-free media containing 10% dialyzed FBS for different time points (0–16 h). Western blot assay was conducted to examine the expression level of BCL2A1 for the above stressful treatments. β-actin was used as a loading control. (**D**) The pcDNA-HA(Ub)8 and pCMV6-BCL2A1-DKK plasmids were transiently transfected to OVCA433 cells. Twenty-four hours after transfection, the cells were treated with MG132 (10 μg) and cultured in hypoxia (0.5% O_2_) or normoxia condition for 16 h. Anti-DDK was used to pull-down ubiquitin-HA. The immunoblotting result showed that hypoxia treatment inhibited the ubiquitin level of BCL2A1 protein. (**E**) Western blot analysis of BCL2A1 in primary ovarian cancer cells isolated from primary ovarian tumors (ovaries), ascites/lavage clinical samples of ovarian cancer patients (P9–P21). β-actin was used as a loading control. (**F**) Quantification of Western blot analysis by gray intensity analysis (Image J) for BCL2A1 expression in primary ovarian cells from primary ovarian tumors (ovaries) and ascites/lavage clinical samples. The expression was normalized to that of β-actin. (**G**) Quantification of Western blot analysis by gray intensity analysis (Image J) for BCL2A1 expression in paired clinical samples (P15 and P21) of primary ovarian tumors and ascites/lavage samples. The expression of BCL2A1 was normalized to that of β-actin. (**H**) BCL2A1 expression in clinical ovarian cancers was examined by immunohistochemical (IHC) analysis using a commercial ovarian cancer tissue array (OV481, US Biomax).

**Figure 4 cancers-13-04577-f004:**
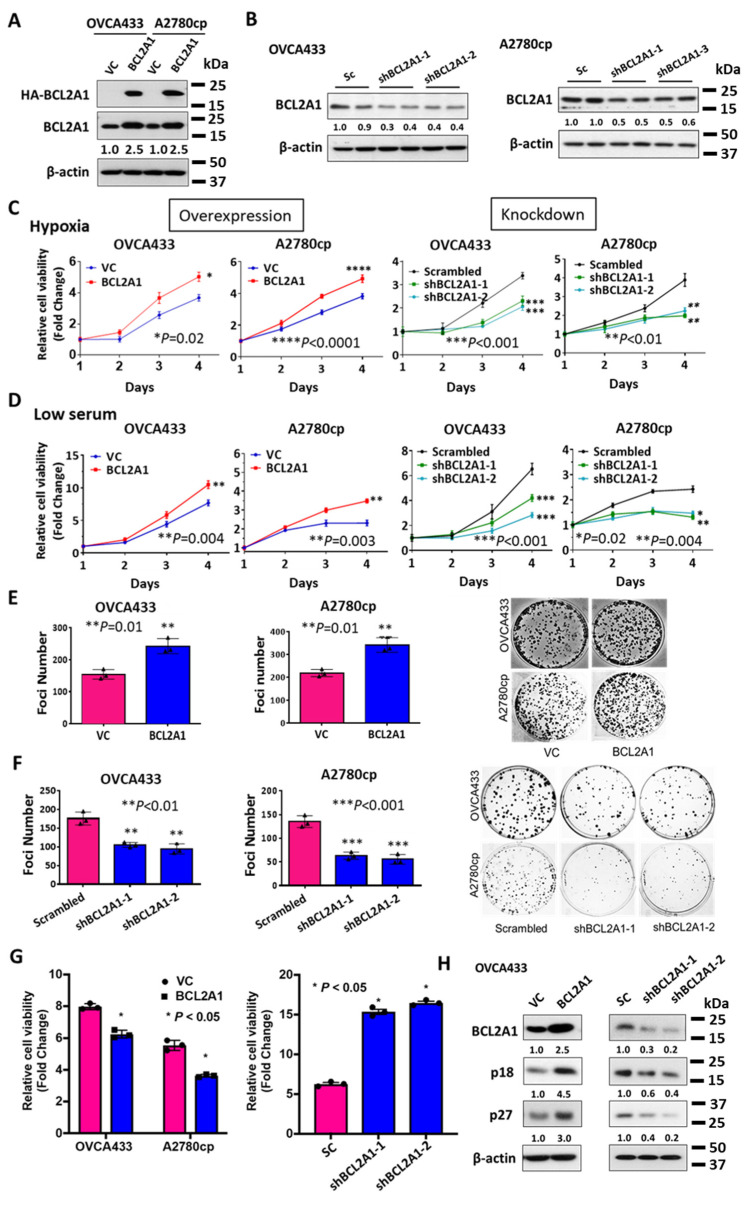
BCL2A1 promotes cell viability under stress. (**A**) Western blotting showed the expression of HA-tagged BCL2A1 in OVCA433 and A2780cp cells. Immunoblotting was performed using anti-HA and ant anti-BCL2A1 antibodies. β-actin was used as an internal control. (**B**) Western blot analysis showed the stable knockdown clones of BCL2A1 mediated by shRNA approach for BCL2A1 in OVCA433 and A2780cp cells. The nonspecific shRNA vector was used as the scrambled control (Sc). (**C**) XTT cell viability assay showed the relative cell availability of the BCL2A1 stably overexpressing clones in OVCA433 (* *p* = 0.02) and A2780cp (**** *p* < 0.0001) cells, as well as stably knockdown clones in OVCA433 (*** *p* < 0.001) and A2780cp (** *p* < 0.01) cells cultured under hypoxic conditions (0.5% O_2_) for four days. (**D**) XTT cell viability assay showed the change of cell viability of BCL2A1 stably overexpressing clones in OVCA433 (** *p* = 0.004) and A2780cp (** *p* = 0.003) cells, as well as stably knockdown clones in OVCA433 (*** *p* < 0.001) and A2780cp (* *p* = 0.02 and ** *p* = 0.004) cells cultured under low serum conditions (0.5% FBS) for four days (** *p* < 0.001). (**E**) Foci formation assay showed the foci formation rate in stable BCL2A1 overexpression or control clones of OVCA433 and A2780cp cells cultured under low serum culture condition (0.5% FBS) (** *p* < 0.01). (**F**) Foci formation assay showed OVCA433 and A2780cp cells with stable BCL2A1 knockdown or control cells cultured under low serum culture conditions (0.5% FBS) (** *p* < 0.01; *** *p* < 0.001). For all, *n* = 3 technical replicates per sample. Data are represented as mean ± S.E.M. (**G**) XTT cell viability assay revealed that enforced expression of BCL2A1 inhibited cell proliferation, whereas knockdown of BCL2A1 enhanced cell proliferation rate of OVCA433 and A2780cp cells in a non-stressed culture on Day 4. Data were represented as mean ± S.E.M (* *p* < 0.05). (**H**) Western blot analysis showed that enforced expression of BCL2A1 induced a higher p18/p27, whereas knockdown exhibited less p18/p27 expression in OVCA433 cells cultured in a non-stressed condition.

**Figure 5 cancers-13-04577-f005:**
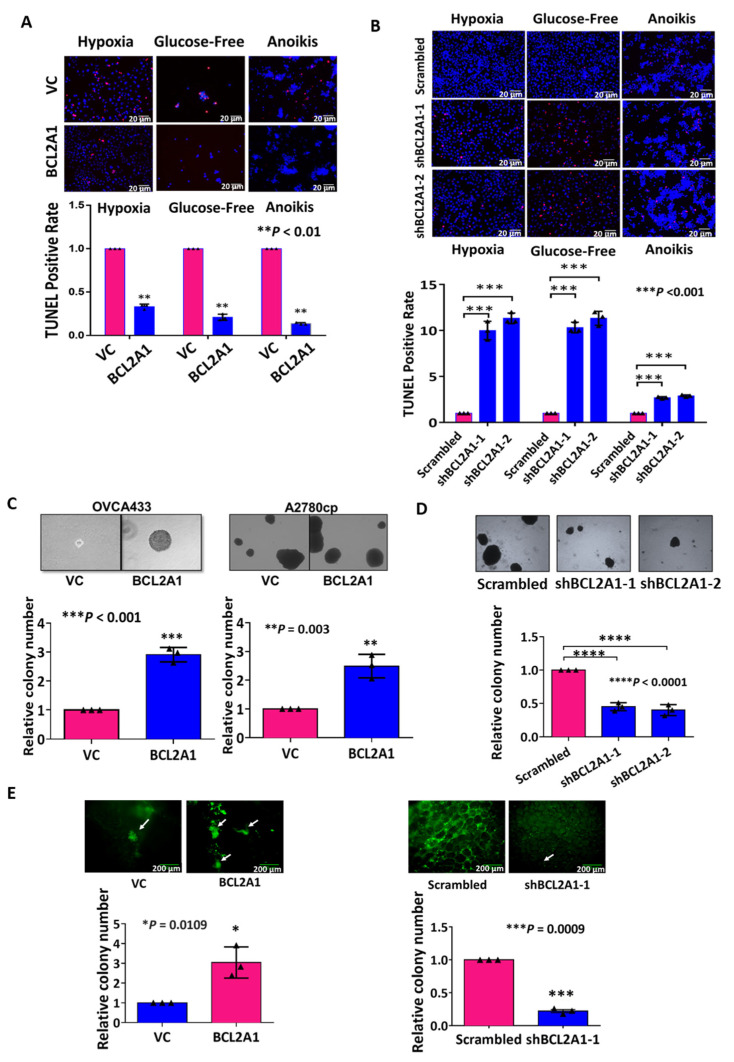
BCL2A1 enhances cell survival and promotes the oncogenic capacities of ovarian cancer cells. TUNEL assay (In Situ Cell Death Detection Kit) revealed the cell apoptosis in OVCA433 cells with (**A**) BCL2A1 overexpression and (**B**) BCL2A1 knockdown cells were cultured under the condition of hypoxia treatment (0.5% O_2_), glucose-fee and anoikis assay for 24–30 h. Images were taken at a magnification of 40×. Scale bar: 20 µm.The bar charts showed the normalized TUNEL positive rates (** *p* < 0.01). VC = empty vector control. (**C**) Soft agar assay showed the anchorage-independent growth ability in BCL2A1 stably overexpressing cells of OVCA433 (*** *p* < 0.001) and A2780cp (** *p* = 0.003). The bar chart showed the relative colony numbers in BCL2A1 overexpressing cells, which was calculated by normalizing the number of the colony of the empty vector control (VC) of each cell line. (**D**) Soft agar assay showed the anchorage-independent growth ability in BCL2A1 knockdown clones (A2780cp-shBCL2A1-1 and A2780cp-shBCL2A1-2) and scrambled control cells (A2780cp-Scrambled) (**** *p* < 0.0001). (**E**) The ex vivo omenta-tumor colonization model showed that the tumor colonization patterns in BCL2A1 stably overexpression clone (BCL2A1) with its empty vector control cells (VC) of GFP-labeled ES-2 cells co-cultured with mice omenta (* *p* = 0.011), and in BCL2A1 stably knockdown clones of GFP-labelled A2780cp (shBCL2A1-1) with its scrambled control (Scrambled) (*** *p* = 0.0009). Tumor colonies (GREEN) were visualized and counted by a fluorescence stereomicroscope (Nikon). Scale bar: 200 µm. For all, *n* = 3 technical replicates per sample. Data were represented as mean ± S.E.M.

**Figure 6 cancers-13-04577-f006:**
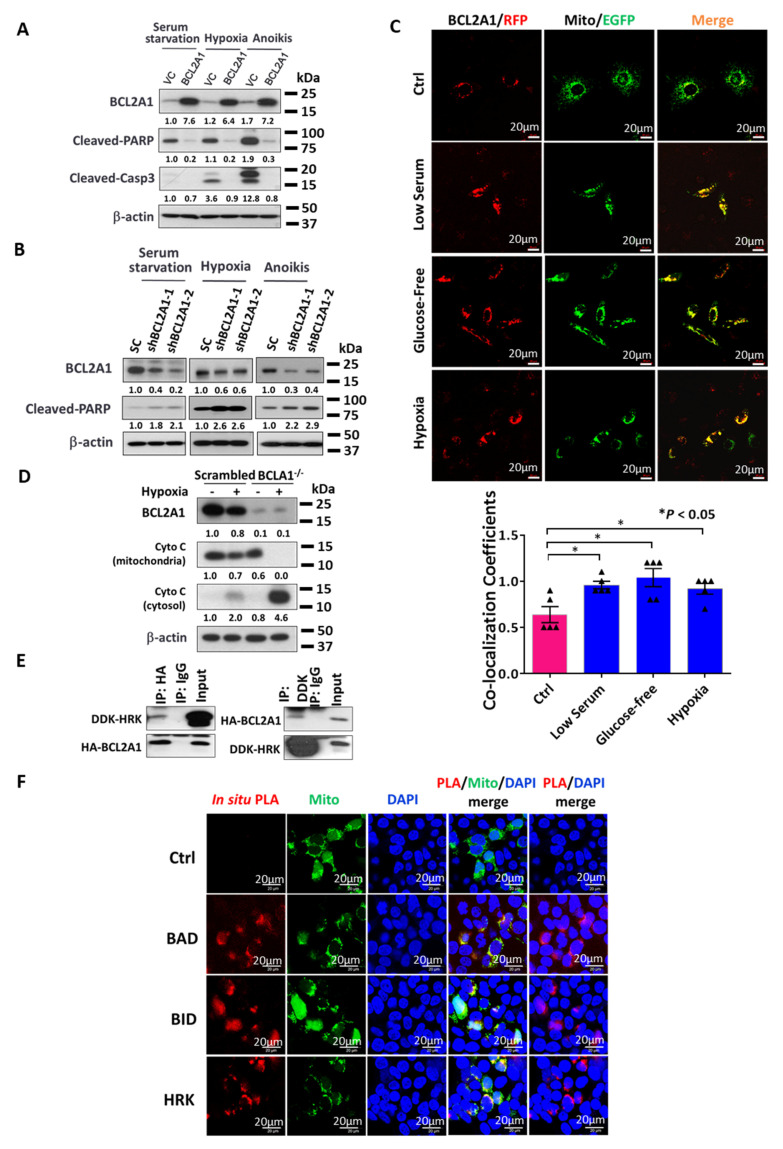
BCL2A1 enhances the cell survival of ovarian cancer cells through suppression of the intrinsic apoptotic pathway. Western blotting analyses revealed that (**A**) overexpression of BCL2A1 reduced, or (**B**) BCL2A1 knockdown increased the PARP and Caspase-3 (Casp3) cleavages in A2780cp and OVCA433 cells, respectively, cultured induced by serum starvation, hypoxia (0.5% O_2_), and anoikis-mediated physiological stresses. (**C**) Representative confocal images show the immunofluorescent staining of BCL2A1-N-RFP (Red), mitochondrial PDHA1-C-GFP (Green) (mitochondria marker, Origene) and DAPI (Blue) in OVCA433 cells transiently co-transfected with BCL2A1-N-RFP and PDHA1-C-GFP and cultured under various stress conditions: Ctrl (normal condition), Low serum medium (0.5% FBS), Glucose-free medium and Hypoxia (0.5% O_2_). Scale bar: 20 μm. The bar chart shows the co-localization efficiency analysis of BCL2A1 in mitochondria by the software ZEN 2012. (* *p* < 0.05 vs Ctrl). (**D**) BCL2A1 knockout elevated the cytochrome C release from mitochondria to cytoplasm under hypoxia culture (0.5% O_2_) for 12 h. (**E**) CoIP assay detected that BCL2A1 interacts with HRK in OVCA433. Immunoblot against DDK showed that DDK existed in the anti-HA immunoprecipitated products, while no blot was observed in the normal IgG pull-down products (Left). Immunoblot against HA demonstrated that HA existed in the anti-DDK pull-down products, while no signal was detected in the normal IgG pull-down products (Right). (**F**) OVCA433 cells were transfected with DDK-tagged expression vector of BCL2A1 and mitochondria targeting GFP expression vector. The in situ PLA staining analysis was performed after 48 h. Co-focal microscopy (40×) was used to detect the signal of protein interaction (Red), mitochondria (Green), and nucleus (Blue). The merged yellow signals (column 4 in row 2–4) demonstrated that the protein binding was mainly localized in the mitochondria. Scale bar: 20 µm. For all, *n* = 3 technical replicates per sample. Data are represented as mean ± S.E.M.

**Figure 7 cancers-13-04577-f007:**
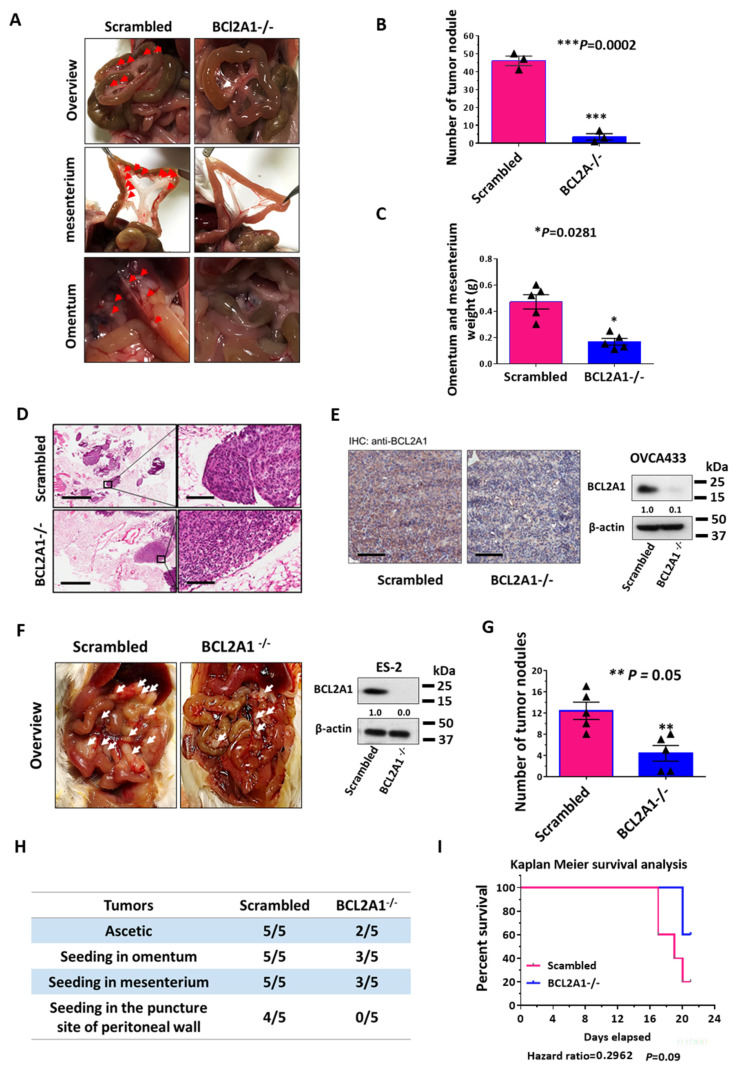
BCL2A1 promotes in vivo tumor dissemination of metastatic ovarian cancer cells. (**A**) The stereomicroscope examination showed that the number of tumor nodules formed by the OVCA433 cells with BCL2A1^−/−^ was much less than those formed by the scrambled control cells. The milky semi-translucent modules highlighted with white arrows represented the tumor nodules formed in mesenturium (upper and middle) and omentum (lower). (**B**) The comparison of the number of tumor nodules in scrambled control and BCL2A1^−/−^ OVCA433 cells. (*** *p* = 0.0002). (**C**) The comparison of the total tumor weight obtained in mesenterium and omentum of OVCA433 with BCL2A1^−/−^ and scrambled OVCA433 cells. (* *p* = 0.0281). (**D**). HE staining of the excised tumor tissues from omentum and mesenterium. The large dark purple spots represented the tumor nodules seeded in the omentum and mesenterium. Cells with large hollows were fat cells. The small homogenous pink-colored spots were the blood vessels of the omentum and mesenterium. (Scale bar: 1 mm (left) and 100 μm (right)). (**E**) IHC staining confirmed that BCL2A1was barely expressed in the tumor nodules that were formed by BCL2A1^−/−^ OVCA433 cells (Scale bar: 100 μm). (**F**) Representative pictures showed tumor nodules formed by BCL2A1^−/−^ ES-2 cells were much less than that of the ES-2 scrambled control (left). The levels of BCL2A1 of ES-2 BCL2A1^−/−^ and scrambled control were confirmed by Western blotting analysis (right). (**G**) The comparison of the number of tumor nodules in scrambled control and BCL2A1^−/−^ ES-2 cells. (** *p* = 0.05). (**H**) The table shows the distribution of tumor spheroids or nodules in the peritoneal cavities of ES-2 BCL2A1^−/−^ and scrambled control. (**I**) Kaplan–Meier survival analysis showed that the mice injected with ES-2 BCL2A1^−/−^ cells had a higher survival probability than those with ES-2 control cells (*p* = 0.09, Log Rank test).

## Data Availability

The microarray data for gynecological cells upon hypoxia treatment has been deposited in ArrayExpress (accession E-MTAB-9730) (http://www.ebi.ac.uk/arrayexpress/experiments/E-MTAB-9730). All data generated or analyzed during this study are included in this published article.

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
