# Peer review of "The Stress-Inducible BCL2A1 Is Required for Ovarian Cancer Metastatic Progression in the Peritoneal Microenvironment"

_cancers, 2021, doi:10.3390/cancers13184577_

Round 1

Reviewer 1 Report

In the manuscript ‘The stress-inducible BCL2A1 is required for ovarian cancer metastatic progression in the peritoneal microenvironment’, the authors found BCL2A1 is a stress induced factor in transcriptome profiling analysis. By using cell models, the authors showed that BCL2A1 enhances ovarian cancer cell proliferation, migration and colony formation in vitro, and tumor metastasis in vivo. The finding is novel. However, before acceptance for publication, some points should be addressed.

Major points:

  1. In figure 1, the authors examined BCL2A1 expression under hypoxia by q-RT-PCR in 3 cell lines, while western blot examination was only run in 2 of the 3 cell lines, and CoCL2 stimulation was only run in in other 2 of the 3 cell lines. In the later part of the manuscript, the authors used cell lines OVCA433 and A2780cp. I wonder why the authors used different cell lines in different sections of the manuscript? I suggest the authors to remove ES-2 in figure 1 since no later experiments were run in this cell line and run CoCL2 stimulation in A2780cp to replace ES-2 in figure 1E. If similar results can’t be repeated in A2780cp, I would doubt the conclusion of the manuscript.
  2. In figure 3D, the authors examined BCL2A1 ubiquitination under hypoxia versus normoxia using MG132 to inhibit proteasome, I think it is necessary to show how it looks like without MG132 inhibition. Furthermore, in HA blot, the authors claim 50 KD and 25 KD bands as tetra- and mono-ubiquitin respectively. However, they might be the heavy and light chain of IgG used for IP but have nothing to do with ubiquitination.
  3. In figure 4C, 4D and 4F, the authors examined cell proliferation and colony formation under stress situation when BCL2A1 was over expressed or its expression was knocked down without examination under non-stress situation. Which leaves suspect that BCL2A1 might do the same thing under non-stress situation. The authors should do the examination under non-stress situation as comparison, otherwise, the authors can’t claim that BCL2A1 is required for cell viability only under stress, the current conclusion will make others consider that BCL2A1 is NOT required for cell viability under non-stress situation.

Similarly, in the later part of the manuscript (figure 5 and 6), the authors did experiments only under stress situation without non-stress situation as control. Which gives the same problem as that in figure 4.

Without comparison the results between non-stress situation and stress situation, the current conclusion can’t be supported.

Minor points:

  1. In line 23, I guess the word ence might be Hence. In line 40, is mor-tality mortality? In line 42, cancer is can-cer? There are numerous similar problems in the manuscript. In line 180 and 181, several symbol @ might be misused. In line 217, 5X105 should be the later 5 should be superscript. Please carefully check the word spelling of the whole manuscript.
  2. Is it necessary to indicate that ovarian cancer cell lines as female source? Are HEK293 cells also female derived?
  3. Part of the figure 1 resolution is too low.
  4. In line 417, the authors used word gain- or loss-of-function. In fact, BCL2A1 was over expressed or its expression was knocked down in the manuscript. Gain- or loss-of-function can be induced by only one amino acid change in the protein without altering the expression level.

Author Response

Major points:

In figure 1, the authors examined BCL2A1 expression under hypoxia by q-RT-PCR in 3 cell lines, while western blot examination was only run in 2 of the 3 cell lines, and CoCL2 stimulation was only run in in other 2 of the 3 cell lines. In the later part of the manuscript, the authors used cell lines OVCA433 and A2780cp. I wonder why the authors used different cell lines in different sections of the manuscript? I suggest the authors to remove ES-2 in figure 1 since no later experiments were run in this cell line and run CoCL2 stimulation in A2780cp to replace ES-2 in figure 1E. If similar results can’t be repeated in A2780cp, I would doubt the conclusion of the manuscript.

Response: As journals request, we need at least more than 2 cell lines as cell models to prove our hypothesis and conclusions. In this study, we used 3 OvCa cell lines for delineating the mechanisms and functions of BCL2A1. In each experiment, we used at least 2 cell lines to draw the conclusion. Importantly, we have provided at least more than one assay for each finding to draw the conclusion. However, due to the limitation of the paper space and the restriction of functional assays, we could not present the results for all 3 cell lines simultaneously. For example, in functional studies, we needed to select some highly motile cell lines for the transwell cell migration and invasion assays or highly tumorigenic for in vivo model. This is commonly accepted in most studies.

In figure 3D, the authors examined BCL2A1 ubiquitination under hypoxia versus normoxia using MG132 to inhibit proteasome, I think it is necessary to show how it looks like without MG132 inhibition. Furthermore, in HA blot, the authors claim 50 KD and 25 KD bands as tetra- and mono-ubiquitin respectively. However, they might be the heavy and light chain of IgG used for IP but have nothing to do with ubiquitination.

Response: BCL2A1 has a dichotomous effect on ovarian cancer in that it inhibits cell growth under normoxia, but it stimulates proliferation and dissemination of ovarian cancer cells under stress. Additionally, it was hypothesized that the induction of the BCL2A1 protein by stresses might undergo post-transcriptional modifications such as ubiquitin-proteasome degradation. Our findings aligned with this hypothesis that the induction of BCL2A1 quickly declined once the prolonged exposure to stresses was adapted, indicating an auto-regulation system of BCL2A1 expression in ovarian cancer cells. Thus, in vivo ubiquitination assay was used to validate this phenomenon, and MG132 proteasome inhibitor should be included in all treatments of the assay to avoid the false-positive effect from rapid degradation. On the other hand, in the assay, the expression pattern of 50 and 25 kDa bands were consistent with the n-ubiquitin and di-ubiquitin upon different treatments, highly suggesting that they are the tetra- and mono-ubiquitin but not the residual antibodies.

In figure 4C, 4D and 4F, the authors examined cell proliferation and colony formation under stress situation when BCL2A1 was over expressed or its expression was knocked down without examination under non-stress situation. Which leaves suspect that BCL2A1 might do the same thing under non-stress situation. The authors should do the examination under non-stress situation as comparison, otherwise, the authors can’t claim that BCL2A1 is required for cell viability only under stress, the current conclusion will make others consider that BCL2A1 is NOT required for cell viability under non-stress situation.

Response: Thanks for the reviewer’s suggestion. We have added the results about the effect on cell survival or proliferation of OvCa which were either stably overexpression or knockdown of BCL2A1 when coculture in non-stress culture conditions (Fig. 4G & 4H) (Supplementary Fig. S5). We found that stably overexpressed BCL2A1 could cause a G1 phase cell cycle arrest by the elevation of p18(INK4c) and p27(Kip1) and vice versa in OvCa cells from nonstressed cultures.

Minor points:

In line 23, I guess the word ence might be Hence. In line 40, is mor-tality mortality? In line 42, cancer is can-cer? There are numerous similar problems in the manuscript. In line 180 and 181, several symbol @ might be misused. In line 217, 5X105 should be the later 5 should be superscript. Please carefully check the word spelling of the whole manuscript.

Response: We found that many words have been altered when formating in the template of this journal. We have amended them accordingly.

Is it necessary to indicate that ovarian cancer cell lines as the female source? Are HEK293 cells also female derived?

Response: We have removed the source of each cell line.

Part of the figure 1 resolution is too low.

Response: We have improved the resolution of each figure.

In line 417, the authors used word gain- or loss-of-function. In fact, BCL2A1 was over expressed or its expression was knocked down in the manuscript. Gain- or loss-of-function can be induced by only one amino acid change in the protein without altering the expression level.

Response: Thanks for point out. We have amended and used “overexpression” and “gene knockdown” to replace word gain- or loss-of-function.

Reviewer 2 Report

The paper by Liang et al, describe the role of a BCL2 protein, BCL2A1, in regulating ovarian cancer progression in hypoxia.

Due to the increasing importance of BCL2 proteins in regulating several aspects of cell biology, this kind of work is necessary to better understand the biology behind these proteins.

The paper is well written and a good amount of experimental work has been done, in order to elucidate the role of this BCL2 protein. For this, the authors have done a good job.

However, some points need to be addressed.

Major comments:

The figures are very low quality, please improve.

The materials and methods section lacks of details describing cell culture under hipoxia. Please improve this. Would be interesting to see how BCL2A1 regulates cellular growth and metastatic potential when cells are constantly growth in hypoxic conditions, since cancer cells grow in oxygen poor environments.

What are the relationships between BCL2A1 and other anti-apoptotic BCL2 proteins such BCL2 and BCL-xL? Since other anti-apoptotic proteins such as BCL2 and BCL-xL are overexpressed in several cancers, seems odd to me that only one BCL2 protein is doing all the work.

It has been described that other anti-apoptotic proteins such as BCL2, BCL-xL and Mcl-1 regulates mitochondrial bioenergetics and cancer growth (also in low oxygen tension). What are the contribution of BCL2A1 in this context? Would be interesting to see if BCL2A1 overexpression contributes to mitochondrial bioenergetics in ovarian cancer in hypoxia.

Some of the BH3 only interaction experiments should be validated in cells with mutations in BH3-only proteins.

Minor comments:

Line 40: change mor-tality to mortality. Throughout the text there are several words with the same problem.

Line 52: could the authors please specify the amount of oxygen present or regulating the ovarian microenvironment ?

Molecular weight Markers for WB are missing in all WB images. Please provide.

CoCl2 is used as an inducer of Hif-1 alfa. The authors should include this information in figure 1E

Line 314. Please correct the word induction

In figure 2C the authors describe the correlation between BCL2A1 and NF-Kb. However, no cuantification of this correlation is actually provided. Please correct this.

Figure 4. the sh RNA does not seem to knowckdown significantly BCL2A1. can the authors provide statistical analysis of quantifications?

Throughout the manuscript, quantification of WB images are provided as numbers. Could the authors provide the SEM or SD of such quantifications as well?

Author Response

Major points:

The figures are very low quality, please improve.

Response: We have improved the resolution of each figure.

The materials and methods section lacks of details describing cell culture under hipoxia. Please improve this. Would be interesting to see how BCL2A1 regulates cellular growth and metastatic potential when cells are constantly growth in hypoxic conditions, since cancer cells grow in oxygen poor environments.

Response:  Thanks for the suggestion of the reviewer. We have added the related information in the sections of Materials and Methods (Luciferase reporter assays) and Supplementary Materials. OvCa cells were treated by hypoxia (0.5% O2) in an Incubator Subchamber (BioSpherix Ltd, Parish, NY, USA).

What are the relationships between BCL2A1 and other anti-apoptotic BCL2 proteins such BCL2 and BCL-xL? Since other anti-apoptotic proteins such as BCL2 and BCL-xL are overexpressed in several cancers, seems odd to me that only one BCL2 protein is doing all the work. It has been described that other anti-apoptotic proteins such as BCL2, BCL-xL and Mcl-1 regulate mitochondrial bioenergetics and cancer growth (also in low oxygen tension). What are the contribution of BCL2A1 in this context? Would be interesting to see if BCL2A1 overexpression contributes to mitochondrial bioenergetics in ovarian cancer in hypoxia.

Response: Overexpression of any of the pro-survival BCL-2 family members prevents the death induced by many apoptotic stimuli, indicating a significant functional redundancy between these proteins. In physiological conditions, it is rare that a single pro-survival protein ensures the survival of a cell population. As an example, T cells successively use BCL-2 and MCL-1, followed by A1 and BCL-XL, then MCL-1 and BCL-XL, and finally BCL-2 again to survive the different stages of their development (1,2). This may reflect either a selective expression of these genes in response to survival signals that may be different at each stage of maturation or an adaptation toward the need to counter apoptotic signals that come in different forms at each of these stages. Genetic studies have helped to identify the physiological role of the pro-survival proteins in mice. The various phenotypes illustrate the different requirements for these proteins in different cell types and at various stages of development (2).

Some of the BH3 only interaction experiments should be validated in cells with mutations in BH3-only proteins.

Response: BH3-only proteins such as BID is an activator of the mitochondrial apoptosis pathway, which directly interacts with and activates BAK/BAX. In fact, BH3-only activators are usually kept by pro-survival proteins such as BCL2A1 until displaced by sensitizer BH3-only proteins such as BAD. Free activators can then bind to and activate the effector protein BAK/BAX and trigger their ologomerization. When mutation occurred in the BH-3 only proteins, not only activators might not be kept by BCL2A1, but also the binding and oligomerization between the free activators displaced and the effector protein (BAK/BAX) might lose. By that time, it is not just the case of whether BCL2A1 and BID sequestration per se can inactivate BAK/BAX to avoid mitochondrial outer membrane per-meabilization and the release of cytochrome c. Thus, further investigations are needed to confirm the effects of BCL2A1 in the sequestration of BAD/HRK and BID with a mutation in the BH3 domain in inducing mitochondrial outer membrane permeabilization.

Minor comments:

Line 40: change mor-tality to mortality. Throughout the text there are several words with the same problem.

Response: We found that many words have been altered when formating in the template of this journal. We have amended them accordingly.

Line 52: could the authors please specify the amount of oxygen present or regulating the ovarian microenvironment ?

Response: We haven’t checked the oxygen content in TME of ovarian cancer in this study. However, according to numerous studies, hypoxia and the activation of HIF1a are often evidenced in TME facilitating cancer development and progression in various solid tumors, including ovarian cancers (Petrova V. et al, Oncogenesis , 2018; Klemba A. et al., Int. J Mol. Sci., 2020)

Molecular weight Markers for WB are missing in all WB images. Please provide.

Response: The instructions of Cancers seem not to request this.

CoCl2 is used as an inducer of Hif-1 alfa. The authors should include this information in figure 1E

Response: We have added “CoCl2, a chemical inducer of HIF1a,….” In the figure legend.

Line 314. Please correct the word induction

Response: We have amended accordingly.

In figure 2C the authors describe the correlation between BCL2A1 and NF-Kb. However, no cuantification of this correlation is actually provided. Please correct this.

Response: We have amended accordingly.

Figure 4. the sh RNA does not seem to knowckdown significantly BCL2A1. can the authors provide statistical analysis of quantifications?

Response: We found that it is challenging to provide a statitical analysis of the WB result. This WB has shown the reduction of BCL2A1 protein level by their relative band intensities.

Throughout the manuscript, quantification of WB images are provided as numbers. Could the authors provide the SEM or SD of such quantifications as well?

Response: The WB presentation has followed the instructions of Cancers.

References

  1. Strasser, A., Puthalakath, H., O'Reilly, L. A., and Bouillet, P. (2008) What do we know about the mechanisms of elimination of autoreactive T and B cells and what challenges remain. Immunol Cell Biol 86, 57-66
  2. Giam, M., Huang, D. C., and Bouillet, P. (2008) BH3-only proteins and their roles in programmed cell death. Oncogene 27 Suppl 1, S128-136

Round 2

Reviewer 1 Report

I think the manuscript is much improved and I suggest for acceptance of the manuscript for publication.

Author Response

We have submitted the certificate of paper editing (by AJE) to the editor via a cover letter.

Reviewer 2 Report

The authors responded to most of the queries.

However, few elements still need to be corrected.

I have asked the authors to add molecular weights marks to the western blot images.

I understand the journal does not require it. However, the answer is not whether the journal requires this. It is good research practice to add such info to these images. 

I have also asked previously:

"In figure 2C the authors describe the correlation between BCL2A1 and NF-Kb. However, no quantification of this correlation is actually provided. Please correct this."

The authors responded that this issue was amended. However, I can't find such modification. Again, please provide such quantification.

Author Response

  1. Comments: The authors responded to most of the queries. However, few elements still need to be corrected. I have asked the authors to add molecular weights marks to the western blot images. I understand the journal does not require it. However, the answer is not whether the journal requires this. It is good research practice to add such info to these images. 

Response: Thanks for the suggestion. We have added the molecular markers accordingly.

  1. Comments: I have also asked previously: "In figure 2C the authors describe the correlation between BCL2A1 and NF-kB. However, no quantification of this correlation is provided. Please correct this.". The authors responded that this issue was amended. However, I can't find such modification. Again, please provide such quantification.

Response: Figure 2C displayed the Western blot analysis on the expression pattern between BCL2A1 and NF-kB in a panel of ovarian cancer cell lines. The purpose of this figure is to overview the expression patterns of BCL2A1 and NF-kB. This western blot doesn’t use for showing any correlation between BCL2A1 expression and the activity of NFkB. The band intensities of these Western blots have been quantified and labelled according to the guidelines of CANCERS. If the reviewer means the correlation of BCL2A1 and NF-kB signaling activities, the results have been shown in Figure 2D-2F.